# Variations on the Expectation Due to Changes in the Probability Measure

**DOI:** 10.3390/e27080865

**Published:** 2025-08-14

**Authors:** Samir M. Perlaza, Gaetan Bisson

**Affiliations:** 1Centre Inria d’Université Côte d’Azur, INRIA, 06902 Sophia Antipolis, France; 2Department of Electrical and Computer Engineering, Princeton University, Princeton, NJ 08544, USA; 3GAATI Mathematics Laboratory, University of French Polynesia, 98702 Faaa, French Polynesia

**Keywords:** Gibbs probability measures, Pythagorean identities, sensitivity, probability distribution drifts, variations of the expectation

## Abstract

In this paper, closed-form expressions for the variation of the expectation of a given function due to changes in the probability measure (probability distribution drifts) are presented. These expressions unveil interesting connections with Gibbs probability measures, information projections, Pythagorean identities for relative entropy, mutual information, and lautum information.

## 1. Introduction

Let *m* be a positive integer and denote by $▵(R^{m})$ the set of all probability measures on the measurable space $\left(R^{m},B\left(R^{m}\right)\right)$, with $B\left(R^{m}\right)$ being the Borel σ-algebra on $R^{m}$. Given a Borel measurable function $h:R^{n}\times R^{m}\to R$, consider the functional(1)$$\begin{array}{c} G_{h}:\left\{\begin{array}{ccc} R^{n}\times ▵\left(R^{m}\right)\times ▵\left(R^{m}\right) & ⟶ & R\\ \left(x,P_{1},P_{2}\right) & ⟼ & \int h\left(x,y\right)dP_{1}\left(y\right)-\int h\left(x,y\right)dP_{2}\left(y\right). \end{array}\right. \end{array}$$ The functional $G_{h}$ in (Equation 1) is defined when both integrals exist and are finite. Hence,(2)$$\begin{array}{ccc} G_{h}\left(x,P_{1},P_{2}\right) & = & \int h\left(x,y\right)dP_{1}\left(y\right)-\int h\left(x,y\right)dP_{2}\left(y\right), \end{array}$$
is the variation of the expectation of the measurable function *h* due to a change in the probability measure from $P_{2}$ to $P_{1}$. These changes in the probability measure are often referred to as probability distribution drifts in some application areas. See for instance [1,2,3] and references therein.

In order to define the expectation of $G_{h}\left(x,P_{1},P_{2}\right)$ in (Equation 2) when *x* is obtained by sampling a probability measure in $▵(R^{n})$, the structure formalized below is required.

**Definition** **1.**
*A family $P_{Y|X}≜{\left(P_{Y|X=x}\right)}_{x\in R^{n}}$ of elements of $▵(R^{m})$ indexed by $R^{n}$ is said to be a conditional probability measure if, for all sets $A\in B\left(R^{m}\right)$, the map*

$$\begin{array}{c} \left\{\begin{array}{ccc} R^{n} & ⟶ & \left[0,1\right]\\ x & ⟼ & P_{Y|X=x}\left(A\right) \end{array}\right. \end{array}$$

*is Borel measurable. The set of all such conditional probability measures is denoted by $▵\left(R^{m}|R^{n}\right)$.*


In this setting, consider the functional(3)$$\begin{array}{c} {\bar{G}}_{h}:\left\{\begin{array}{ccc} ▵\left(R^{m}|R^{n}\right)\times ▵\left(R^{m}|R^{n}\right)\times ▵\left(R^{n}\right) & ⟶ & R\\ \left(P_{Y|X}^{\left(1\right)},P_{Y|X}^{\left(2\right)},P_{X}\right) & ⟼ & \int G_{h}\left(x,P_{Y|X=x}^{\left(1\right)},P_{Y|X=x}^{\left(2\right)}\right)dP_{X}\left(x\right) \end{array}\right.. \end{array}$$ Hence,(4)$$\begin{array}{ccc} {\bar{G}}_{h}\left(P_{Y|X}^{\left(1\right)},P_{Y|X}^{\left(2\right)},P_{X}\right) & = & \int G_{h}\left(x,P_{Y|X=x}^{\left(1\right)},P_{Y|X=x}^{\left(2\right)}\right)dP_{X}\left(x\right) \end{array}$$(5)$$\begin{array}{ccc} & = & \int \left(\int h\left(x,y\right)dP_{Y|X=x}^{\left(1\right)}\left(y\right)-\int h\left(x,y\right)dP_{Y|X=x}^{\left(2\right)}\left(y\right)\right)dP_{X}\left(x\right) \end{array}$$(6)$$\begin{array}{ccc} & = & \int h\left(x,y\right)dP_{Y|X}^{\left(1\right)}P_{X}\left(y,x\right)-\int h\left(x,y\right)dP_{Y|X}^{\left(2\right)}P_{X}\left(y,x\right), \end{array}$$
where the functional $G_{h}$ is defined in (Equation 1), is the variation of the integral (expectation) of the function *h* when the probability measure changes from the joint probability measure $P_{Y|X}^{\left(1\right)}P_{X}$ to another joint probability measure $P_{Y|X}^{\left(2\right)}P_{X}$, both in $▵\left(R^{m}\times R^{n}\right)$.

Special attention is given to the quantity ${\bar{G}}_{h}\left(P_{Y},P_{Y|X},P_{X}\right)$, for some $P_{Y|X}\in ▵\left(R^{m}|R^{n}\right)$, with $P_{Y}$ being the marginal on $▵\left(R^{m}\right)$ of the joint probability measure $P_{Y|X}P_{X}$ on $▵\left(R^{m}\times R^{n}\right)$. That is, for all sets $A\in B\left(R^{m}\right)$,(7)$$\begin{array}{ccc} P_{Y}\left(A\right) & = & \int P_{Y|X=x}\left(A\right)dP_{X}\left(x\right). \end{array}$$ The relevance of the quantity ${\bar{G}}_{h}\left(P_{Y},P_{Y|X},P_{X}\right)$ stems from the fact that it captures the variation of the expectation of the function *h* when the probability measure changes from the joint probability measure $P_{Y|X}P_{X}$ to the product of its marginals $P_{Y}P_{X}$. That is,(8)$$\begin{array}{ccc} {\bar{G}}_{h}\left(P_{Y},P_{Y|X},P_{X}\right) & = & \int G_{h}\left(x,P_{Y},P_{Y|X=x}\right)dP_{X}\left(x\right) \end{array}$$(9)$$\begin{array}{ccc} & = & \int \left(\int h\left(x,y\right)dP_{Y}\left(y\right)-\int h\left(x,y\right)dP_{Y|X=x}\left(y\right)\right)dP_{X}\left(x\right) \end{array}$$(10)$$\begin{array}{ccc} & = & \int h\left(x,y\right)dP_{Y}P_{X}\left(y,x\right)-\int h\left(x,y\right)dP_{Y|X}P_{X}\left(y,x\right), \end{array}$$
where the functional $G_{h}$ is defined in (Equation 1).

### 1.1. Novelty and Contributions

This work makes two key contributions: First, it provides a closed-form expression for the variation $G_{h}\left(x,P_{1},P_{2}\right)$ in (Equation 2) for a fixed $x\in R^{n}$ and two arbitrary probability measures $P_{1}$ and $P_{2}$, formulated explicitly in terms of relative entropies. Second, it derives a closed-form expression for the expected variation ${\bar{G}}_{h}\left(P_{Y|X}^{\left(1\right)},P_{Y|X}^{\left(2\right)},P_{X}\right)$ in (Equation 4), again in terms of information measures, for arbitrary conditional probability measures $P_{Y|X}^{\left(1\right)}$, $P_{Y|X}^{\left(2\right)}$, and an arbitrary probability measure $P_{X}$.

A further contribution of this work is the derivation of specific closed-form expressions for ${\bar{G}}_{h}\left(P_{Y},P_{Y|X},P_{X}\right)$ in (Equation 10), which reveal deep connections with both mutual information [4] and lautum information [5]. Notably, when $P_{Y|X}$ is a Gibbs conditional probability measure, this variation simplifies (up to a constant factor) to the sum of the mutual and lautum information induced by the joint distribution $P_{Y|X}P_{X}$.

These results were originally discovered in the analysis of generalization error of machine-learning algorithms, see for instance [6,7,8,9,10]. Therein, the function *h* in (Equation 2) was assumed to represent an empirical risk. This paper presents such results in a comprehensive and general setting that is no longer tied to such assumptions. Also, strong connections with information projections and Pythagorean identities [11,12] are discussed. This new general presentation not only unifies previously scattered insights but also makes the results applicable across a broad range of domains in which probability distribution shifts are relevant.

### 1.2. Applications

The study of the variation of the integral (expectation) of *h* (for some fixed $x\in R^{n}$) due to a measure change from $P_{2}$ to $P_{1}$, i.e., the value $G_{h}\left(x,P_{1},P_{2}\right)$ in (Equation 2), plays a central role in the definition of integral probability metrics (IPMs) [13,14]. Using the notation in (Equation 2), an IPM results from the optimization problem(11)$$\begin{array}{c} \underset{h\in H}{\sup}\left|G_{h}\left(x,P_{1},P_{2}\right)\right|, \end{array}$$
for some fixed $x\in R^{n}$ and a particular class of functions H. Note, for instance, that the maximum mean discrepancy [15] and the Wasserstein distance of order one [16,17,18,19] are both IPMs.

Other areas of mathematics in which the variation $G_{h}\left(x,P_{1},P_{2}\right)$ in (Equation 2) plays a key role are distributionally robust optimization (DRO) [20,21] and optimization with relative entropy regularization [7,8]. In these areas, the variation $G_{h}\left(x,P_{1},P_{2}\right)$ is a central tool. See for instance, [6,22] and references therein.

Variations of the form $G_{h}\left(x,P_{1},P_{2}\right)$ in (Equation 2) have also been studied in [9,10] in the particular case of statistical machine learning for the analysis of generalization error. The central observation is that the generalization error of machine-learning algorithms can be written in the form ${\bar{G}}_{h}\left(P_{Y},P_{Y|X},P_{X}\right)$ in (Equation 10). This observation is the main building block of the method of gaps introduced in [10], which leads to a number of closed-form expressions for the generalization error involving mutual information, lautum information, among other information measures.

## 2. Preliminaries

The main results presented in this work involve Gibbs conditional probability measures. Such measures are parametrized by a Borel measurable function $h:R^{n}\times R^{m}\to R$; a σ-finite measure *Q* on $R^{m}$; and a real λ∈R. Often, the measure *Q* is called the reference measure, and the real parameter λ is called the *temperature* parameter. The use of a σ-finite reference measure was first introduced in [7]. This class of measures includes the Lebesgue measure and the counting measure, enabling a unified treatment of random variables with either probability density functions or probability mass functions. The standard case in which the reference measure is a probability measure representing a prior is also covered. Finally, note that the variable *x* will remain inactive until Section 4. However, it is introduced now for consistency.

Consider the following function:(12)$$\begin{array}{c} K_{h,Q,x}:\left\{\begin{array}{ccc} R & ⟶ & R\\ t & ⟼ & \log\left(\int \exp\left(t\;h(x,y)\right)dQ\left(y\right)\right). \end{array}\right. \end{array}$$ Under the assumption that *Q* is a probability measure, the function $K_{h,Q,x}$ in (Equation 12) is the cumulant generating function of the random variable h(x,Y), for some fixed $x\in R^{n}$ and Y∼Q. Using this notation, the definition of the Gibbs conditional probability measure is presented hereunder.

**Definition** **2**(Gibbs Conditional Probability Measure)**.**
*Given a Borel measurable function $h:R^{n}\times R^{m}\to R$; a σ-finite measure Q on $R^{m}$; and a λ∈R, the probability measure $P_{Y|X}^{\left(h,Q,\lambda \right)}\in ▵\left(R^{m}|R^{n}\right)$ is said to be an (h,Q,λ)-Gibbs conditional probability measure if*(13)$$\begin{array}{ccc} \forall x\in X,K_{h,Q,x}\left(-\lambda \right) & < & +\infty ; \end{array}$$
*for some set $X\subseteq R^{n}$; and for all (x,y)∈X×suppQ,*
(14)$$\begin{array}{ccc} \frac{dP_{Y|X=x}^{\left(h,Q,\lambda \right)}}{dQ}\left(y\right) & = & \exp\left(-\lambda h\left(x,y\right)-K_{h,Q,x}\left(-\lambda \right)\right), \end{array}$$
*where the function $K_{h,Q,x}$ is defined in (Equation 12); the set suppQ is the support of the σ-finite measure Q; and the function $\frac{dP_{Y|X=x}^{\left(h,Q,\lambda \right)}}{dQ}$ is the Radon-Nikodym derivative [23,24] of the probability measure $P_{Y|X=x}^{\left(h,Q,\lambda \right)}$ with respect to Q.*

Note that, while $P_{Y|X}^{\left(h,Q,\lambda \right)}$ is an (h,Q,λ)-Gibbs conditional probability measure, the measure $P_{Y|X=x}^{\left(h,Q,\lambda \right)}$, obtained by conditioning it upon a given vector x∈X, is referred to as an (h,Q,λ)-Gibbs probability measure.

The condition in (Equation 13) is easily met under certain assumptions. For instance, if *h* is a nonnegative function and *Q* is a finite measure, then it holds for all $\lambda \in \left(0,+\infty \right)$. Let ${▵}_{Q}\left(R^{m}\right)≜\left\{P\in ▵\left(R^{m}\right):P\ll Q\right\}$, with P≪Q standing for “*P* absolutely continuous with respect to *Q*”. The relevance of (h,Q,λ)-Gibbs probability measures relies on the fact that under some conditions, they are the unique solutions to problems of the form,(15)$$\begin{array}{ccc} & & \min_{P\in {▵}_{Q}\left(R^{m}\right)}\int h\left(x,y\right)dP\left(y\right)+\frac{1}{\lambda }D\left(P∥Q\right),\mathrm{and} \end{array}$$(16)$$\begin{array}{ccc} & & \max_{P\in {▵}_{Q}\left(R^{m}\right)}\int h\left(x,y\right)dP\left(y\right)+\frac{1}{\lambda }D\left(P∥Q\right), \end{array}$$
where λ∈R∖{0}, x∈R, and $D\left(P∥Q\right)$ denotes the relative entropy (or KL divergence) of *P* with respect to *Q*.

**Definition** **3**(Relative Entropy)**.**
*Given two σ-finite measures P and Q on the same measurable space, such that P is absolutely continuous with respect to Q, the relative entropy of P with respect to Q is*(17)$$D\left(P∥Q\right)=\int \frac{dP}{dQ}\left(x\right)\log\left(\frac{dP}{dQ}\left(x\right)\right)dQ\left(x\right),$$
*where the function $\frac{dP}{dQ}$ is the Radon-Nikodym derivative of P with respect to Q.*

The key observation is that when λ>0, the objective function in (Equation 15) is convex with *P*. Alternative, when λ<0, the objective function in (Equation 16) is concave. The connection between the optimization problems (Equation 15) and (Equation 16) and the Gibbs probability measure $P_{Y|X=x}^{\left(h,Q,\lambda \right)}$ in (Equation 14) has been pointed out by several authors. See for instance, Theorem 3 in [7] and [6,25,26,27,28,29,30,31,32,33] for the former; and Theorem 1 in [9], together with [34,35,36] for the latter. In these references, a variety of assumptions and proof techniques have been used to highlight such connections. A general and unified statement of these observations is presented hereunder.

**Lemma** **1.**
*Assume that the optimization problem in (Equation 15) (respectively, in (Equation 16)) admits a solution. Then, if λ>0 (respectively, if λ<0), the probability measure $P_{Y|X=x}^{\left(h,Q,\lambda \right)}$ in (Equation 14) is the unique solution.*


**Proof.** For the case in which λ>0, the proof follows the same approach as the proof of Theorem 3 in [7]. Alternatively, for the case in which λ<0, the proof follows along the lines of the proof of Theorem 1 in [9]. □

The following lemma highlights a key property of (h,Q,λ)-Gibbs probability measures.

**Lemma** **2.**
*Given an (h,Q,λ)-Gibbs probability measure, denoted by $P_{Y|X=x}^{\left(h,Q,\lambda \right)}$, with $x\in R^{n}$,*

(18)
$$\begin{array}{ccc} -\frac{1}{\lambda }K_{h,Q,x}\left(-\lambda \right) & = & \int h\left(x,y\right)dQ\left(y\right)-\frac{1}{\lambda }D\left(Q∥P_{Y|X=x}^{\left(h,Q,\lambda \right)}\right) \end{array}$$


(19)
$$\begin{array}{ccc} & = & \int h\left(x,y\right)dP_{Y|X=x}^{\left(h,Q,\lambda \right)}\left(y\right)+\frac{1}{\lambda }D\left(P_{Y|X=x}^{\left(h,Q,\lambda \right)}∥Q\right); \end{array}$$

*moreover, if λ>0,*

(20)
$$\begin{array}{ccc} -\frac{1}{\lambda }K_{h,Q,x}\left(-\lambda \right) & = & \min_{P\in {▵}_{Q}\left(R^{m}\right)}\int h\left(x,y\right)dP\left(y\right)+\frac{1}{\lambda }D\left(P∥Q\right); \end{array}$$

*alternatively, if λ<0,*

(21)
$$\begin{array}{ccc} -\frac{1}{\lambda }K_{h,Q,x}\left(-\lambda \right) & = & \max_{P\in {▵}_{Q}\left(R^{m}\right)}\int h\left(x,y\right)dP\left(y\right)+\frac{1}{\lambda }D\left(P∥Q\right), \end{array}$$

*where the function $K_{h,Q,x}$ is defined in (Equation 12).*


**Proof.** The proof of (Equation 19) follows from taking the logarithm of both sides of (Equation 14) and integrating with respect to $P_{Y|X=x}^{\left(h,Q,\lambda \right)}$. As for the proof of (Equation 18), it follows by noticing that for all $\left(x,y\right)\in R^{n}\times \mathrm{supp}\;Q$, the Radon-Nikodym derivative $\frac{dP_{Y|X=x}^{\left(h,Q,\lambda \right)}}{dQ}\left(y\right)$ in (Equation 14) is strictly positive. Thus, from Theorem 5 in [37], it holds that $\frac{dQ}{dP_{Y|X=x}^{\left(h,Q,\lambda \right)}}\left(y\right)={\left(\frac{dP_{Y|X=x}^{\left(h,Q,\lambda \right)}}{dQ}\left(y\right)\right)}^{-1}$. Hence, taking the negative logarithm on both sides of (Equation 14) and integrating with respect to *Q* leads to (Equation 18). Finally, the equalities in (Equation 20) and (Equation 21) follow from Lemma 1 and (Equation 19). □

The Equalities (Equation 19)–(Equation 21) in Lemma 2 can be seen as an immediate restatement of Donsker–Varadhan variational representation of the relative entropy [38]. Alternative interesting proofs for (Equation 18) have been presented by several authors including [9,33]. A proof for (Equation 19) appears in [6] (Lemma 3), in the specific case of λ>0.

The following lemma introduces the main building block of this work, which is a characterization of the variation of the expectation of the function $h\left(x,\cdot \right):R^{m}\to R$ when the probability measure changes from the probability measure $P_{Y|X=x}^{\left(h,Q,\lambda \right)}$ in (Equation 14) to an arbitrary measure $P\in {▵}_{Q}\left(R^{m}\right)$, i.e., $G_{h}\left(x,P,P_{Y|X=x}^{\left(h,Q,\lambda \right)}\right)$, for some fixed $x\in R^{n}$. Such a result appeared for the first time in [6] (Theorem 1) for the case in which λ>0; and in [9] (Theorem 6) for the case in which λ<0, in different contexts of statistical machine learning. A general and unified statement of such results is presented hereunder.

**Lemma** **3.**
*Consider an (h,Q,λ)-Gibbs probability measure, denoted by $P_{Y|X=x}^{\left(h,Q,\lambda \right)}\in ▵\left(R^{m}\right)$, with λ≠0 and x∈R. For all $P\in {▵}_{Q}\left(R^{m}\right)$,*

(22)
$$\begin{array}{ccc} G_{h}\left(x,P,P_{Y|X=x}^{\left(h,Q,\lambda \right)}\right) & = & \frac{1}{\lambda }\left(D\left(P∥P_{Y|X=x}^{\left(h,Q,\lambda \right)}\right)+D\left(P_{Y|X=x}^{\left(h,Q,\lambda \right)}∥Q\right)-D\left(P∥Q\right)\right). \end{array}$$



**Proof.** The proof follows along the lines of the proofs of Theorem 1 in [6] and Theorem 6 in [9] for the cases in which λ>0 and λ<0, respectively. A unified proof is presented hereunder by noticing that for all $P\in {▵}_{Q}\left(R^{m}\right)$,(23)$$\begin{array}{ccc} D\left(P∥P_{Y|X=x}^{\left(h,Q,\lambda \right)}\right) & = & \int \log\left(\frac{dP}{dP_{Y|X=x}^{\left(h,Q,\lambda \right)}}\left(y\right)\right)dP\left(y\right) \end{array}$$(24)$$\begin{array}{ccc} & = & \int \log\left(\frac{dQ}{dP_{Y|X=x}^{\left(h,Q,\lambda \right)}}\left(y\right)\frac{dP}{dQ}\left(y\right)\right)dP\left(y\right) \end{array}$$(25)$$\begin{array}{ccc} & = & \int \log\left(\frac{dQ}{dP_{Y|X=x}^{\left(h,Q,\lambda \right)}}\left(y\right)\right)dP\left(y\right)+D\left(P∥Q\right) \end{array}$$(26)$$\begin{array}{ccc} & = & \lambda \int h\left(x,y\right)dP\left(y\right)+K_{h,Q,x}\left(-\lambda \right)+D\left(P∥Q\right) \end{array}$$(27)$$\begin{array}{ccc} & = & \lambda G_{h}\left(x,P,P_{Y|X=x}^{\left(h,Q,\lambda \right)}\right)-D\left(P_{Y|X=x}^{\left(h,Q,\lambda \right)}∥Q\right)+D\left(P∥Q\right), \end{array}$$
where (Equation 24) follows from Theorem 4 in [37]; (Equation 26) follows from Theorem 5 in [37]; and (Equation 14); and (Equation 27) follows from (Equation 19). □

It is interesting to highlight that $G_{h}\left(x,P,P_{Y|X=x}^{\left(h,Q,\lambda \right)}\right)$ in (Equation 22) characterizes the variation of the expectation of the function $h\left(x,\cdot \right):R^{m}\to R$, when λ>0 (respectively, when λ<0) and the probability measure changes from the solution to the optimization problem in (Equation 15) (respectively, in (Equation 16)) to an alternative measure *P*. This result takes another perspective if it is seen in the context of information projections [12]. Let *Q* be a probability measure and $S\subseteq {▵}_{Q}\left(R^{m}\right)$ be a convex set. From Theorem 1 in [12], it holds that for all measures P∈S,(28)$$\begin{array}{c} D\left(P∥Q\right)⩾D\left(P∥P^{★}\right)+D\left(P^{★}∥Q\right), \end{array}$$
where $P^{★}$ satisfies(29)$$\begin{array}{c} P^{★}\in \arg\min_{P\in S}D\left(P∥Q\right). \end{array}$$ In the particular case in which the set S in (Equation 29) satisfies(30)$$\begin{array}{ccc} S & ≜ & \left\{P\in {▵}_{Q}\left(R^{m}\right):\int h\left(x,y\right)dP\left(y\right)=c\right\}, \end{array}$$
for some real *c*, with the vector *x* and the function *h* defined in Lemma 3, the optimal measure $P^{★}$ in (Equation 29) is the Gibbs probability measure $P_{Y|X=x}^{\left(h,Q,\lambda \right)}$ in (Equation 14), with λ>0 chosen to satisfy(31)$$\begin{array}{c} \int h\left(x,y\right)dP_{Y|X=x}^{\left(h,Q,\lambda \right)}\left(y\right)=c. \end{array}$$ The case in which the measure *Q* in (Equation 29) is a σ-finite measure, for instance, either the Lebesgue measure or the counting measure, respectively, leads to the classical framework of differential entropy or discrete entropy maximization, which have been studied under particular assumptions on the set S in [34,35,36].

When the reference measure *Q* is a probability measure, under the assumption that (Equation 31) holds, it follows from Theorem 3 in [12] that for all P∈S, with S in (Equation 30),(32)$$\begin{array}{ccc} D\left(P∥Q\right) & = & D\left(P∥P_{Y|X=x}^{\left(h,Q,\lambda \right)}\right)+D\left(P_{Y|X=x}^{\left(h,Q,\lambda \right)}∥Q\right), \end{array}$$
which is known as the Pythagorean theorem for relative entropy. Such a geometric interpretation follows from admitting relative entropy as an analog of squared Euclidean distance. The first appearance of such a “Pythagorean theorem” was in [11] and was later revisited in [12]. Interestingly, the same result can be obtained from Lemma 3 by noticing that for all P∈S, with S in (Equation 30),(33)$$\begin{array}{ccc} G_{h}\left(x,P,P_{Y|X=x}^{\left(h,Q,\lambda \right)}\right) & = & 0. \end{array}$$

The converse of the Pythagorean theorem, e.g., Proposition 48 in [39], together with Lemma 3, lead to the geometric construction shown in Figure 1. A similar interpretation was also presented in [10] in the context of the generalization error of machine-learning algorithms. Nonetheless, the interpretation in Figure 1 is general and independent of such an application.

The relevance of Lemma 3 in the context of information projections follows from the fact that *Q* might be a σ-finite measure. The class of σ-finite measures includes the class of probability measures, and thus, unifies the results separately obtained in the realm of maximum entropy methods and information-projection methods.

The following lemma highlights that (h,Q,λ)-Gibbs conditional probability measures are related to another class of optimization problems.

**Lemma** **4.**
*Assume that the following optimization problems possess at least one solution for some $x\in R^{n}$,*

(34)
$$\begin{array}{ccc} \min_{P\in {▵}_{Q}\left(R^{m}\right)} & \; & \int h(x,y)dP(y)\\ s.t. & & D\left(P∥Q\right)⩽\rho . \end{array}$$

*and*

(35)
$$\begin{array}{ccc} \max_{P\in {▵}_{Q}\left(R^{m}\right)} & \; & \int h(x,y)dP(y)\\ s.t. & & D\left(P∥Q\right)⩽\rho . \end{array}$$

*Consider the (h,Q,λ)-Gibbs probability measure $P_{Y|X=x}^{\left(h,Q,\lambda \right)}$ in (Equation 14), with λ∈R∖{0} such that $\rho =D\left(P_{Y|X=x}^{\left(h,Q,\lambda \right)}∥Q\right)$. Then, the (h,Q,λ)-Gibbs probability measure $P_{Y|X=x}^{\left(h,Q,\lambda \right)}$ is a solution to (Equation 34) if λ>0; or to (Equation 35) if λ<0.*


**Proof.** Note that if λ>0, then, $\frac{1}{\lambda }D\left(P∥P_{Y|X=x}^{\left(h,Q,\lambda \right)}\right)⩾0$. Hence, from Lemma 3, it holds that for all probability measures *P* such that $D\left(P∥Q\right)⩽\rho$,(36)$$\begin{array}{ccc} G_{h}\left(x,P,P_{Y|X=x}^{\left(h,Q,\lambda \right)}\right) & ⩾ & \frac{1}{\lambda }\left(D\left(P_{Y|X=x}^{\left(h,Q,\lambda \right)}∥Q\right)-D\left(P∥Q\right)\right) \end{array}$$(37)$$\begin{array}{ccc} & = & \frac{1}{\lambda }\left(\rho -D\left(P∥Q\right)\right) \end{array}$$(38)$$\begin{array}{ccc} & ⩾ & 0,\;\;\;\;\;\;\;\;\; \end{array}$$
with equality if $D\left(P∥P_{Y|X=x}^{\left(h,Q,\lambda \right)}\right)=0$. This implies that $P_{Y|X=x}^{\left(h,Q,\lambda \right)}$ is a solution to (Equation 34). Note also that if λ<0, from Lemma 3, it holds that for all probability measures *P* such that $D\left(P∥Q\right)⩽\rho$,(39)$$\begin{array}{ccc} G_{h}\left(x,P,P_{Y|X=x}^{\left(h,Q,\lambda \right)}\right) & ⩽ & \frac{1}{\lambda }\left(D\left(P_{Y|X=x}^{\left(h,Q,\lambda \right)}∥Q\right)-D\left(P∥Q\right)\right) \end{array}$$(40)$$\begin{array}{ccc} & = & \frac{1}{\lambda }\left(\rho -D\left(P∥Q\right)\right) \end{array}$$(41)$$\begin{array}{ccc} & ⩽ & 0,\;\;\;\;\;\;\;\;\; \end{array}$$
with equality if $D\left(P∥P_{Y|X=x}^{\left(h,Q,\lambda \right)}\right)=0$. This implies that $P_{Y|X=x}^{\left(h,Q,\lambda \right)}$ is a solution to (Equation 35). □

## 3. Characterization of $G_{h}\left(x,P_{1},P_{2}\right)$ in (Equation 2)

The main result of this section is the following theorem.

**Theorem** **1.**
*For all probability measures $P_{1}$ and $P_{2}$, both absolutely continuous with respect to a given σ-finite measure Q on $R^{m}$, the variation $G_{h}\left(x,P_{1},P_{2}\right)$ in (Equation 2) satisfies,*

(42)
$$\begin{array}{ccc} G_{h}\left(x,P_{1},P_{2}\right) & = & \frac{1}{\lambda }\left(D\left(P_{1}∥P_{Y|X=x}^{\left(h,Q,\lambda \right)}\right)-D\left(P_{2}∥P_{Y|X=x}^{\left(h,Q,\lambda \right)}\right)+D\left(P_{2}∥Q\right)-D\left(P_{1}∥Q\right)\right), \end{array}$$

*where the probability measure $P_{Y|X=x}^{\left(h,Q,\lambda \right)}$, with λ≠0, is an (h,Q,λ)-Gibbs probability measure.*


**Proof.** The proof follows from Lemma 3 and by observing that$$G_{h}\left(x,P_{1},P_{2}\right)=G_{h}\left(x,P_{1},P_{Y|X=x}^{\left(h,Q,\lambda \right)}\right)-G_{h}\left(x,P_{2},P_{Y|X=x}^{\left(h,Q,\lambda \right)}\right),$$
which completes the proof. □

Theorem 1 might be particularly simplified in the case in which the reference measure *Q* is a probability measure. Consider for instance the case in which $P_{1}\ll P_{2}$ (or $P_{2}\ll P_{1}$). In such a case, the reference measure might be chosen as $P_{2}$ (or $P_{1}$), as shown hereunder.

**Corollary** **1.**
*Consider the variation $G_{h}\left(x,P_{1},P_{2}\right)$ in (Equation 2). If the probability measure $P_{1}$ is absolutely continuous with respect to $P_{2}$, then,*

(43)
$$\begin{array}{ccc} G_{h}\left(x,P_{1},P_{2}\right) & = & \frac{1}{\lambda }\left(D\left(P_{1}∥P_{Y|X=x}^{\left(h,P_{2},\lambda \right)}\right)-D\left(P_{2}∥P_{Y|X=x}^{\left(h,P_{2},\lambda \right)}\right)-D\left(P_{1}∥P_{2}\right)\right). \end{array}$$

*Alternatively, if the probability measure $P_{2}$ is absolutely continuous with respect to $P_{1}$, then,*

(44)
$$\begin{array}{ccc} G_{h}\left(x,P_{1},P_{2}\right) & = & \frac{1}{\lambda }\left(D\left(P_{1}∥P_{Y|X=x}^{\left(h,P_{1},\lambda \right)}\right)-D\left(P_{2}∥P_{Y|X=x}^{\left(h,P_{1},\lambda \right)}\right)+D\left(P_{2}∥P_{1}\right)\right), \end{array}$$

*where the probability measures $P_{Y|X=x}^{\left(h,P_{1},\lambda \right)}$ and $P_{Y|X=x}^{\left(h,P_{2},\lambda \right)}$ are, respectively, $\left(h,P_{1},\lambda \right)$- and $\left(h,P_{2},\lambda \right)$-Gibbs probability measures, with λ≠0.*


In the case in which neither $P_{1}$ is absolutely continuous with respect to $P_{2}$; nor $P_{2}$ is absolutely continuous with respect to $P_{1}$, the reference measure *Q* in Theorem 1 can always be chosen as a convex combination of $P_{1}$ and $P_{2}$. That is, for all Borel sets $A\in B\left(R^{m}\right)$, $Q\left(A\right)=\alpha P_{1}\left(A\right)+\left(1-\alpha \right)P_{2}\left(A\right)$, with α∈(0,1).

Theorem 1 can be specialized to the cases in which *Q* is either the Lebesgue measure or the counting measure.

If *Q* is the Lebesgue measure, then the probability measures $P_{1}$ and $P_{2}$ in (Equation 42) admit probability density functions $f_{1}$ and $f_{2}$, respectively. Moreover, the terms $-D\left(P_{1}∥Q\right)$ and $-D\left(P_{2}∥Q\right)$ are Shannon’s differential entropies [4] induced by $P_{1}$ and $P_{2}$, denoted by $h(P_{1})$ and $h(P_{2})$, respectively. That is, for all i∈{1,2},(45)$$\begin{array}{ccc} h(P_{i}) & ≜ & -\int f_{i}\left(x\right)\log f_{i}\left(x\right)dx. \end{array}$$ The probability measure $P_{Y|X=x}^{\left(h,Q,\lambda \right)}$, with λ≠0, $x\in R^{n}$, and *Q* the Lebesgue measure, possesses a probability density function, denoted by $f_{Y|X=x}^{\left(h,Q,\lambda \right)}:R^{m}\to \left(0,+\infty \right)$, which satisfies(46)$$\begin{array}{ccc} f_{Y|X=x}^{\left(h,Q,\lambda \right)}\left(y\right) & = & \frac{\exp\left(-\lambda h(x,y)\right)}{\int \exp\left(-\lambda h(x,y)\right)dy}. \end{array}$$

If *Q* is the counting measure, then the probability measures $P_{1}$ and $P_{2}$ in (Equation 42) admit probability mass functions $p_{1}:Y\to \left[0,1\right]$ and $p_{2}:Y\to \left[0,1\right]$, with Y a countable subset of $R^{m}$. Moreover, $-D\left(P_{1}∥Q\right)$ and $-D\left(P_{2}∥Q\right)$ are, respectively, Shannon’s discrete entropies [4] induced by $P_{1}$ and $P_{2}$, denoted by $H(P_{1})$ and $H(P_{2})$, respectively. That is, for all i∈{1,2},(47)$$\begin{array}{ccc} H(P_{i}) & ≜ & -\sum_{y\in Y}p_{i}\left(y\right)\log p_{i}\left(y\right). \end{array}$$ The probability measure $P_{Y|X=x}^{\left(h,Q,\lambda \right)}$, with λ≠0 and *Q* the counting measure, possesses a probability mass function, denoted by $p_{Y|X=x}^{\left(h,Q,\lambda \right)}:Y\to \left(0,+\infty \right)$, which satisfies(48)$$\begin{array}{ccc} p_{Y|X=x}^{\left(h,Q,\lambda \right)}\left(y\right) & = & \frac{\exp\left(-\lambda h(x,y)\right)}{\sum_{y\in Y}\exp\left(-\lambda h(x,y)\right)}. \end{array}$$

These observations lead to the following corollary of Theorem 1.

**Corollary** **2.**
*Given two probability measures $P_{1}$ and $P_{2}$, with probability density functions $f_{1}$ and $f_{2}$, respectively, the variation $G_{h}\left(x,P_{1},P_{2}\right)$ in (Equation 2) satisfies,*

(49)
$$\begin{array}{ccc} G_{h}\left(x,P_{1},P_{2}\right) & = & \frac{1}{\lambda }\left(D\left(P_{1}∥P_{Y|X=x}^{\left(h,Q,\lambda \right)}\right)-D\left(P_{2}∥P_{Y|X=x}^{\left(h,Q,\lambda \right)}\right)-h\left(P_{2}\right)+h\left(P_{1}\right)\right),\; \end{array}$$

*where the probability density function of the measure $P_{Y|X=x}^{\left(h,Q,\lambda \right)}$, with λ≠0 and Q the Lebesgue measure, is defined in (Equation 46); and the entropy functional h is defined in (Equation 45). Alternatively, given two probability measures $P_{1}$ and $P_{2}$, with probability mass functions $p_{1}$ and $p_{2}$, respectively, the variation $G_{h}\left(x,P_{1},P_{2}\right)$ in (Equation 2) satisfies,*

(50)
$$\begin{array}{ccc} G_{h}\left(x,P_{1},P_{2}\right) & = & \frac{1}{\lambda }\left(D\left(P_{1}∥P_{Y|X=x}^{\left(h,Q,\lambda \right)}\right)-D\left(P_{2}∥P_{Y|X=x}^{\left(h,Q,\lambda \right)}\right)-H\left(P_{2}\right)+H\left(P_{1}\right)\right),\; \end{array}$$

*where the probability mass function of the measure $P_{Y|X=x}^{\left(h,Q,\lambda \right)}$, with λ≠0 and Q the counting measure, is defined in (Equation 48); and the entropy functional H is defined in (Equation 47).*


## 4. Characterizations of ${\bar{G}}_{h}\left(P_{Y|X}^{\left(1\right)},P_{Y|X}^{\left(2\right)},P_{X}\right)$ in (Equation 4)

The main result of this section is a characterization of ${\bar{G}}_{h}\left(P_{Y|X}^{\left(1\right)},P_{Y|X}^{\left(2\right)},P_{X}\right)$ in (Equation 4).

**Theorem** **2.**
*Consider the variation ${\bar{G}}_{h}\left(P_{Y|X}^{\left(1\right)},P_{Y|X}^{\left(2\right)},P_{X}\right)$ in (Equation 4) and assume that for all $x\in \mathrm{supp}\;P_{X}$, the probability measures $P_{Y|X=x}^{\left(1\right)}$ and $P_{Y|X=x}^{\left(2\right)}$ are both absolutely continuous with respect to a σ-measure Q. Then,*

(51)
$$\begin{array}{ccc} {\bar{G}}_{h}\left(P_{Y|X}^{\left(1\right)},P_{Y|X}^{\left(2\right)},P_{X}\right) & = & \frac{1}{\lambda }\int (D\left(P_{Y|X=x}^{\left(1\right)}∥P_{Y|X=x}^{\left(h,Q,\lambda \right)}\right)-D\left(P_{Y|X=x}^{\left(2\right)}∥P_{Y|X=x}^{\left(h,Q,\lambda \right)}\right)\\ & & +D\left(P_{Y|X=x}^{\left(2\right)}∥Q\right)-D\left(P_{Y|X=x}^{\left(1\right)}∥Q\right))dP_{X}\left(x\right), \end{array}$$

*where the probability measure $P_{Y|X}^{\left(h,Q,\lambda \right)}$, with λ≠0, is an (h,Q,λ)-Gibbs conditional probability measure.*


**Proof.** The proof follows from (Equation 4) and Theorem 1. □

Two special cases are particularly noteworthy. When the reference measure *Q* is the Lebesgue measure both $-\int D\left(P_{Y|X=x}^{\left(1\right)}∥Q\right)dP_{X}\left(x\right)$ and $-\int D\left(P_{Y|X=x}^{\left(2\right)}∥Q\right)dP_{X}\left(x\right)$ in (Equation 51) become Shannon’s differential conditional entropies, denoted by $h\left(P_{Y|X}^{\left(1\right)}|P_{X}\right)$ and $h\left(P_{Y|X}^{\left(2\right)}|P_{X}\right)$, respectively. That is, for all i∈{1,2},(52)$$\begin{array}{ccc} h\left(P_{Y|X}^{\left(i\right)}|P_{X}\right) & ≜ & \int h\left(P_{Y|X=x}^{\left(i\right)}\right)dP_{X}\left(x\right), \end{array}$$
where h is the entropy functional in (Equation 45).

When the reference measure *Q* is the counting measure both $-\int D\left(P_{Y|X=x}^{\left(1\right)}∥Q\right)dP_{X}\left(x\right)$ and $-\int D\left(P_{Y|X=x}^{\left(2\right)}∥Q\right)dP_{X}\left(x\right)$ in (Equation 51) become Shannon’s discrete conditional entropies, denoted by $H\left(P_{Y|X}^{\left(1\right)}|P_{X}\right)$ and $H\left(P_{Y|X}^{\left(2\right)}|P_{X}\right)$, respectively. That is, for all i∈{1,2},(53)$$\begin{array}{ccc} H\left(P_{Y|X}^{\left(i\right)}|P_{X}\right) & ≜ & \int H\left(P_{Y|X=x}^{\left(i\right)}\right)dP_{X}\left(x\right), \end{array}$$
where H is the entropy functional in (Equation 47).

These observations lead to the following corollary of Theorem 2.

**Corollary** **3.**
*Consider the variation ${\bar{G}}_{h}\left(P_{Y|X}^{\left(1\right)},P_{Y|X}^{\left(2\right)},P_{X}\right)$ in (Equation 4) and assume that for all $x\in \mathrm{supp}\;P_{X}$, the probability measures $P_{Y|X=x}^{\left(1\right)}$ and $P_{Y|X=x}^{\left(2\right)}$ possess probability density functions. Then,*

(54)
$$\begin{array}{ccc} {\bar{G}}_{h}\left(P_{Y|X}^{\left(1\right)},P_{Y|X}^{\left(2\right)},P_{X}\right) & = & \frac{1}{\lambda }\int \left(D\left(P_{Y|X=x}^{\left(1\right)}∥P_{Y|X=x}^{\left(h,Q,\lambda \right)}\right)-D\left(P_{Y|X=x}^{\left(2\right)}∥P_{Y|X=x}^{\left(h,Q,\lambda \right)}\right)\right)dP_{X}\left(x\right)\\ & & -\frac{1}{\lambda }h\left(P_{Y|X}^{\left(2\right)}|P_{X}\right)+\frac{1}{\lambda }h\left(P_{Y|X}^{\left(1\right)}|P_{X}\right), \end{array}$$

*where the probability density function of the measure $P_{Y|X=x}^{\left(h,Q,\lambda \right)}$, with λ≠0 and Q the Lebesgue measure, is defined in (Equation 46); and for all i∈{1,2}, the conditional entropy $h\left(P_{Y|X}^{\left(i\right)}|P_{X}\right)$ is defined in (Equation 52). Alternatively, assume that for all $x\in \mathrm{supp}\;P_{X}$, the probability measures $P_{Y|X=x}^{\left(1\right)}$ and $P_{Y|X=x}^{\left(2\right)}$ possess probability mass functions. Then,*

(55)
$$\begin{array}{ccc} {\bar{G}}_{h}\left(P_{Y|X}^{\left(1\right)},P_{Y|X}^{\left(2\right)},P_{X}\right) & = & \frac{1}{\lambda }\int \left(D\left(P_{Y|X=x}^{\left(1\right)}∥P_{Y|X=x}^{\left(h,Q,\lambda \right)}\right)-D\left(P_{Y|X=x}^{\left(2\right)}∥P_{Y|X=x}^{\left(h,Q,\lambda \right)}\right)\right)dP_{X}\left(x\right)\\ & & -\frac{1}{\lambda }H\left(P_{Y|X}^{\left(2\right)}|P_{X}\right)+\frac{1}{\lambda }H\left(P_{Y|X}^{\left(1\right)}|P_{X}\right), \end{array}$$

*where the probability mass function of the measure $P_{Y|X=x}^{\left(h,Q,\lambda \right)}$, with λ≠0 and Q the counting measure, is defined in (Equation 48); and for all i∈{1,2}, the conditional entropy $H\left(P_{Y|X}^{\left(i\right)}|P_{X}\right)$ is defined in (Equation 53).*


The general expression for the expected variation ${\bar{G}}_{h}\left(P_{Y|X}^{\left(1\right)},P_{Y|X}^{\left(2\right)},P_{X}\right)$ in (Equation 4) might be simplified according to Corollary 1. For instance, if for all $x\in \mathrm{supp}\;P_{X}$, the probability measure $P_{Y|X=x}^{\left(1\right)}$ is absolutely continuous with respect to $P_{Y|X=x}^{\left(2\right)}$, the measure $P_{Y|X=x}^{\left(2\right)}$ can be chosen to be the reference measure in the calculation of $G_{h}\left(x,P_{Y|X=x}^{\left(1\right)},P_{Y|X=x}^{\left(2\right)}\right)$, with the functional $G_{h}$ in (Equation 1). This observation leads to the following corollary of Theorem 2.

**Corollary** **4.**
*Consider the variation ${\bar{G}}_{h}\left(P_{Y|X}^{\left(1\right)},P_{Y|X}^{\left(2\right)},P_{X}\right)$ in (Equation 4) and assume that for all $x\in \mathrm{supp}\;P_{X}$, $P_{Y|X=x}^{\left(1\right)}\ll P_{Y|X=x}^{\left(2\right)}$. Then,*

(56)
$$\begin{array}{ccc} {\bar{G}}_{h}\left(P_{Y|X}^{\left(1\right)},P_{Y|X}^{\left(2\right)},P_{X}\right) & = & \frac{1}{\lambda }\int (D\left(P_{Y|X=x}^{\left(1\right)}∥P_{Y|X=x}^{\left(h,P_{Y|X=x}^{\left(2\right)},\lambda \right)}\right)-D\left(P_{Y|X=x}^{\left(2\right)}∥P_{Y|X=x}^{\left(h,P_{Y|X=x}^{\left(2\right)},\lambda \right)}\right)\\ & & -D\left(P_{Y|X=x}^{\left(1\right)}∥P_{Y|X=x}^{\left(2\right)}\right))dP_{X}\left(x\right). \end{array}$$

*Alternatively, if for all $x\in \mathrm{supp}\;P_{X}$, the probability measure $P_{Y|X=x}^{\left(2\right)}$ is absolutely continuous with respect to $P_{Y|X=x}^{\left(1\right)}$, then,*

(57)
$$\begin{array}{ccc} {\bar{G}}_{h}\left(P_{Y|X}^{\left(1\right)},P_{Y|X}^{\left(2\right)},P_{X}\right) & = & \frac{1}{\lambda }\int (D\left(P_{Y|X=x}^{\left(1\right)}∥P_{Y|X=x}^{\left(h,P_{Y|X=x}^{\left(1\right)},\lambda \right)}\right)-D\left(P_{Y|X=x}^{\left(2\right)}∥P_{Y|X=x}^{\left(h,P_{Y|X=x}^{\left(1\right)},\lambda \right)}\right)\\ & & +D\left(P_{Y|X=x}^{\left(2\right)}∥P_{Y|X=x}^{\left(1\right)}\right))dP_{X}\left(x\right), \end{array}$$

*where the measures $P_{Y|X=x}^{\left(h,P_{Y|X=x}^{\left(1\right)},\lambda \right)}$ and $P_{Y|X=x}^{\left(h,P_{Y|X=x}^{\left(2\right)},\lambda \right)}$ are, respectively, $\left(h,P_{Y|X=x}^{\left(1\right)},\lambda \right)$- and (h, $P_{Y|X=x}^{\left(2\right)},\lambda )$-Gibbs probability measures.*


The Gibbs probability measures $P_{Y|X=x}^{\left(h,P_{Y|X=x}^{\left(1\right)},\lambda \right)}$ and $P_{Y|X=x}^{\left(h,P_{Y|X=x}^{\left(2\right)},\lambda \right)}$ in Corollary 4 are particularly interesting as their reference measures depend on *x*. Gibbs measures of this form appear, for instance, in Corollary 10 in [7].

## 5. Characterizations of ${\bar{G}}_{h}\left(P_{Y},P_{Y|X},P_{X}\right)$ in (Equation 10)

The main result of this section is a characterization of ${\bar{G}}_{h}\left(P_{Y},P_{Y|X},P_{X}\right)$ in (Equation 10), which describes the variation of the expectation of the function *h* when the probability measure changes from the joint probability measure $P_{Y|X}P_{X}$ to the product of its marginals $P_{Y}P_{X}$.

This result is presented hereunder and involves the mutual information $I\left(P_{Y|X};P_{X}\right)$ and lautum information $L\left(P_{Y|X};P_{X}\right)$, defined as follows:(58)$$\begin{array}{ccc} I\left(P_{Y|X};P_{X}\right) & ≜ & \int D\left(P_{Y|X=x}∥P_{Y}\right)dP_{X}\left(x\right);\mathrm{and} \end{array}$$(59)$$\begin{array}{ccc} L\left(P_{Y|X};P_{X}\right) & ≜ & \int D\left(P_{Y}∥P_{Y|X=x}\right)dP_{X}\left(x\right). \end{array}$$

**Theorem** **3.**
*Consider the expected variation ${\bar{G}}_{h}\left(P_{Y},P_{Y|X},P_{X}\right)$ in (Equation 10) and assume that, for all $x\in \mathrm{supp}\;P_{X}$:*
*1.* 
*The probability measures $P_{Y}$ and $P_{Y|X=x}$ are both absolutely continuous with respect to a given σ-finite measure Q; and*
*2.* 
*The probability measures $P_{Y}$ and $P_{Y|X=x}$ are mutually absolutely continuous.*

*Then,*

(60)
$$\begin{array}{ccc} & & {\bar{G}}_{h}\left(P_{Y},P_{Y|X},P_{X}\right)=\frac{1}{\lambda }(I\left(P_{Y|X};P_{X}\right)+L\left(P_{Y|X};P_{X}\right)\\ & & +\;\int \int \;\log\left(\frac{dP_{Y|X=x}}{dP_{Y|X=x}^{\left(h,Q,\lambda \right)}}\left(y\right)\right)dP_{Y}\left(y\right)dP_{X}\left(x\right)-\;\int \int \;\log\left(\frac{dP_{Y|X=x}}{dP_{Y|X=x}^{\left(h,Q,\lambda \right)}}\left(y\right)\right)dP_{Y|X=x}\left(y\right)dP_{X}\left(x\right)), \end{array}$$

*where the probability measure $P_{Y|X}^{\left(h,Q,\lambda \right)}$, with λ≠0, is an (h,Q,λ)-Gibbs conditional probability measure.*


**Proof.** The proof is presented in Appendix A. □

An alternative expression for ${\bar{G}}_{h}\left(P_{Y},P_{Y|X},P_{X}\right)$ in (Equation 10) involving only relative entropies is presented by the following theorem.

**Theorem** **4.**
*Consider the expected variation ${\bar{G}}_{h}\left(P_{Y},P_{Y|X},P_{X}\right)$ in (Equation 10) and assume that, for all $x\in \mathrm{supp}\;P_{X}$, the probability measure $P_{Y|X=x}$ is absolutely continuous with respect to a given σ-finite measure Q. Then, it follows that*

(61)
$$\begin{array}{ccc} & & {\bar{G}}_{h}\left(P_{Y},P_{Y|X},P_{X}\right)\\ & = & \frac{1}{\lambda }\;\int \int \;\left(D\left(P_{Y|X=x_{2}}∥P_{Y|X=x_{1}}^{\left(h,Q,\lambda \right)}\right)-D\left(P_{Y|X=x_{2}}∥P_{Y|X=x_{2}}^{\left(h,Q,\lambda \right)}\right)\right)dP_{X}\left(x_{1}\right)dP_{X}\left(x_{2}\right), \end{array}$$

*where $P_{Y|X}^{\left(h,Q,\lambda \right)}$, with λ≠0, is an (h,Q,λ)-Gibbs conditional probability measure.*


**Proof.** The proof is presented in Appendix B. □

Theorem 4 expresses the variation ${\bar{G}}_{h}\left(P_{Y},P_{Y|X},P_{X}\right)$ in (Equation 10) as difference of two relative entropies. The former compares $P_{Y|X=x_{1}}$ with $P_{Y|X=x_{2}}^{\left(h,Q,\lambda \right)}$, where $(x_{1},x_{2})\in X\times X$ are independently sampled from the same probability measure $P_{X}$. The latter compares these two conditional measures conditioning on the same element of X. That is, it compares $P_{Y|X=x_{2}}$ with $P_{Y|X=x_{2}}^{\left(h,Q,\lambda \right)}$.

An interesting observation from Theorems 3 and 4 is that the last two terms in the right-hand side of (Equation 60) are both zero in the case in which $P_{Y|X}$ is an (h,Q,λ)-Gibbs conditional probability measure. Similarly, in such a case, the second term in the right-hand side of (Equation 61) is also zero. This observation is highlighted by the following corollary.

**Corollary** **5.**
*Consider an (h,Q,λ)-Gibbs conditional probability measure, denoted by $P_{Y|X}^{\left(h,Q,\lambda \right)}\in ▵\left(R^{m}|R^{n}\right)$, with λ≠0; and a probability measure $P_{X}\in ▵\left(R^{n}\right)$. Let the measure $P_{Y}^{\left(h,Q,\lambda \right)}\in ▵\left(R^{m}\right)$ be such that for all sets $A\in B\left(R^{m}\right)$,*

(62)
$$\begin{array}{ccc} P_{Y}^{\left(h,Q,\lambda \right)}\left(A\right) & = & \int P_{Y|X=x}^{\left(h,Q,\lambda \right)}\left(A\right)dP_{X}\left(x\right). \end{array}$$

*Then,*

(63)
$$\begin{array}{ccc} {\bar{G}}_{h}\left(P_{Y}^{\left(h,Q,\lambda \right)},P_{Y|X}^{\left(h,Q,\lambda \right)},P_{X}\right) & = & \frac{1}{\lambda }\left(I\left(P_{Y|X}^{\left(h,Q,\lambda \right)};P_{X}\right)+L\left(P_{Y|X}^{\left(h,Q,\lambda \right)};P_{X}\right)\right) \end{array}$$


(64)
$$\begin{array}{ccc} & = & \frac{1}{\lambda }\;\int \int \;D\left(P_{Y|X=x_{2}}^{\left(h,Q,\lambda \right)}∥P_{Y|X=x_{1}}^{\left(h,Q,\lambda \right)}\right)dP_{X}\left(x_{1}\right)dP_{X}\left(x_{2}\right). \end{array}$$



Please note that mutual information and lautum information are both nonnegative information measures, which from Corollary 5, implies that ${\bar{G}}_{h}\left(P_{Y}^{\left(h,Q,\lambda \right)},P_{Y|X}^{\left(h,Q,\lambda \right)},P_{X}\right)$ in (Equation 64) might be either positive or negative depending exclusively on the sign of the regularization factor λ. The following corollary exploits such an observation to present a property of Gibbs conditional probability measures and their corresponding marginal probability measures.

**Corollary** **6.**
*Given a probability measure $P_{X}\in ▵\left(R^{n}\right)$, the (h,Q,λ)-Gibbs conditional probability measure $P_{Y|X}^{\left(h,Q,\lambda \right)}$ in (Equation 14) and the probability measure $P_{Y}^{\left(h,Q,\lambda \right)}$ in (Equation 62) satisfy*

(65)
$$\begin{array}{ccc} \;\int \int \;h\left(x,y\right)dP_{Y}^{\left(h,Q,\lambda \right)}\left(y\right)dP_{X}\left(x\right) & ⩾ & \;\int \int \;h\left(x,y\right)dP_{Y|X=x}^{\left(h,Q,\lambda \right)}\left(y\right)dP_{X}\left(x\right)\;if\;\lambda >0; \end{array}$$

*or*

(66)
$$\begin{array}{ccc} \;\int \int \;h\left(x,y\right)dP_{Y}^{\left(h,Q,\lambda \right)}\left(y\right)dP_{X}\left(x\right) & ⩽ & \;\int \int \;h\left(x,y\right)dP_{Y|X=x}^{\left(h,Q,\lambda \right)}\left(y\right)dP_{X}\left(x\right)\;if\;\lambda <0. \end{array}$$



Corollary 6 highlights the fact that a deviation from the joint probability measure $P_{Y|X}^{\left(h,Q,\lambda \right)}P_{X}\in ▵\left(Y\times X\right)$ to the product of its marginals $P_{Y}^{\left(h,Q,\lambda \right)}P_{X}\in ▵\left(Y\times X\right)$ might increase or decrease the expectation of the function *h* depending on the sign of λ.

## 6. Examples

An immediate application of the results presented above is the analysis of the generalization error of machine-learning algorithms [10], which was the scenario in which these results were originally discovered. In the remainder of this section, such results are presented as consequences of the more general results presented in this work.

Let M, X and Y, with $M\subseteq R^{d}$, be sets of *models*, *patterns*, and *labels*, respectively. A pair (x,y)∈X×Y is referred to as a *data point*. A dataset $z\in {\left(X\times Y\right)}^{n}$ is a tuple of *n* data points of the form:(67)$$z=\left(\left(x_{1},y_{1}\right),\left(x_{2},y_{2}\right),\ldots ,\left(x_{n},y_{n}\right)\right)\in {\left(X\times Y\right)}^{n}.$$ Consider the function(68)$$\begin{array}{c} L:\left\{\begin{array}{ccc} {\left(X\times Y\right)}^{n}\times M & ⟶ & \left[0,+\infty \right]\\ \left(z,\theta \right) & ⟼ & \frac{1}{n}\sum_{i=1}^{n}\ell \left(x_{i},y_{i},\theta \right), \end{array}\right. \end{array}$$
where $\ell \left(x_{i},y_{i},\theta \right)$ is the risk or loss induced by the model θ with respect to the data point $\left(x_{i},y_{i}\right)$.

Given a fixed dataset z of the form in (Equation 67), consider also the functional(69)$$R_{z}:\left\{\begin{array}{ccc} ▵\left(M\right) & ⟶ & R\\ P & ⟼ & \int L\left(z,\theta \right)dP\left(\theta \right), \end{array}\right.$$
where the function L is defined in (Equation 68). Using this notation, the *empirical risk* induced by the model θ with respect to the dataset z is $L\left(z,\theta \right)$. The expectation of the empirical risk with respect to a fixed dataset z when models are sampled from a probability measure $P\in ▵\left(M\right)$ is $R_{z}\left(P\right)$.

A machine-learning algorithm is represented by a conditional probability measure $P_{\Theta |Z}\in ▵\left(M|{\left(X\times Y\right)}^{n}\right)$. The instance of such an algorithm generated by training it upon the dataset z in (Equation 67) is represented by the probability measure $P_{\Theta |Z=z}\in ▵\left(M\right)$. The generalization error induced by the algorithm $P_{\Theta |Z}$ is defined as follows.

**Definition** **4**(Generalization Error)**.**
*The generalization error induced by the algorithm $P_{\Theta |Z}\in ▵\left(M|{\left(X\times Y\right)}^{n}\right)$, under the assumption that training and test datasets are independently sampled from a probability measure $P_{Z}\in ▵\left({\left(X\times Y\right)}^{n}\right)$, is denoted by $\bar{\bar{G}}\left(P_{\Theta |Z},P_{Z}\right)$, and*(70)$$\begin{array}{ccc} \bar{\bar{G}}\left(P_{\Theta |Z},P_{Z}\right) & ≜ & \int \int \left(R_{u}\left(P_{\Theta |Z=z}\right)-R_{z}\left(P_{\Theta |Z=z}\right)\right)dP_{Z}\left(u\right)dP_{Z}\left(z\right), \end{array}$$
*where the functionals $R_{u}$ and $R_{z}$ are defined in (Equation 69).*

Often, the term $R_{u}\left(P_{\Theta |Z=z}\right)$ is recognized to be the test error induced by the algorithm $P_{\Theta |Z}$ with respect to a test dataset $u\in {\left(X\times Y\right)}^{n}$ when it is trained upon the dataset $z\in {\left(X\times Y\right)}^{n}$. Alternatively, the term $R_{z}\left(P_{\Theta |Z=z}\right)$ is recognized to be the training error induced by the algorithm $P_{\Theta |Z}$ when it is trained upon the dataset $z\in {\left(X\times Y\right)}^{n}$. From this perspective, the generalization error $\bar{\bar{G}}\left(P_{\Theta |Z},P_{Z}\right)$ in (Equation 70) is the expectation of the difference between test error and training error when the test and training datasets are independently sampled from the same probability measure $P_{Z}$. The key observation is that such generalization error $\bar{\bar{G}}\left(P_{\Theta |Z},P_{Z}\right)$ can be written as a variation of an expectation of the empirical risk function L in (Equation 68), as shown hereunder.

**Lemma** **5**(Lemma 3 in [10])**.**
*Consider the generalization error $\bar{\bar{G}}\left(P_{\Theta |Z},P_{Z}\right)$ in (Equation 70) and assume that for all z, the probability measure $P_{\Theta |Z=z}$ is absolutely continuous with respect to the probability measure $P_{\Theta }\in ▵\left(M\right)$, which satisfies for all measurable subsets C of M,*(71)$$\begin{array}{ccc} P_{\Theta }\left(C\right) & = & \int P_{\Theta |Z=z}\left(C\right)dP_{Z}\left(z\right). \end{array}$$
*Then,*
(72)$$\begin{array}{ccc} \bar{\bar{G}}\left(P_{\Theta |Z},P_{Z}\right) & = & {\bar{G}}_{L}\left(P_{\Theta },P_{\Theta |Z},P_{Z}\right), \end{array}$$
*where the functional ${\bar{G}}_{L}$ and the function L are defined in (Equation 3) and (Equation 68), respectively.*

From Theorems 3 and 5, the following holds.

**Theorem** **5**(Theorem 14 in [10])**.**
*Consider the generalization error $\bar{\bar{G}}\left(P_{\Theta |Z},P_{Z}\right)$ in (Equation 70) and assume that for all $z\in {\left(X\times Y\right)}^{n}$:*
*(a)* *The probability measures $P_{\Theta }$ in (Equation 71) and $P_{\Theta |Z=z}$ are both absolutely continuous with respect to some σ-finite measure $Q\in ▵\left(M\right)$;**(b)* *The measure Q is absolutely continuous with respect to $P_{\Theta }$; and**(c)* *The measure $P_{\Theta }$ is absolutely continuous with respect to $P_{\Theta |Z=z}$.*
*Then,*
(73)$$\begin{array}{ccc} \bar{\bar{G}}\left(P_{\Theta |Z},P_{Z}\right) & = & \lambda \left(I\left(P_{\Theta |Z};P_{Z}\right)+L\left(P_{\Theta |Z};P_{Z}\right)\right)+\lambda \int \int \log\frac{dP_{\Theta |Z=z}}{dP_{\Theta |Z=z}^{\left(Q,\lambda \right)}}\left(\theta \right)dP_{\Theta }\left(\theta \right)dP_{Z}\left(z\right)\\ & & -\lambda \int \int \log\frac{dP_{\Theta |Z=z}}{dP_{\Theta |Z=z}^{\left(Q,\lambda \right)}}\left(\theta \right)dP_{\Theta |Z=z}\left(\theta \right)dP_{Z}\left(z\right), \end{array}$$
*where the measure $P_{\Theta |Z}^{\left(Q,\lambda \right)}$ is an (L,Q,λ)-Gibbs conditional probability measure, with the function L defined in (Equation 68).*

Theorem 5 shows one of many closed-form expressions that can be obtained for the generalization error $\bar{\bar{G}}\left(P_{\Theta |Z},P_{Z}\right)$ in (Equation 70) in terms of information measures. A complete exposition of several equivalent alternative expressions, as well as a discussion on their relevance, is presented in [10].

The important observation in this example is that the measure $P_{\Theta |Z}^{\left(Q,\lambda \right)}$ in (Equation 73) is an (L,Q,λ)-Gibbs conditional probability measure, which represents the celebrated *Gibbs algorithm* in statistical machine learning [40]. Thus, the term $\log\frac{dP_{\Theta |Z=z}}{dP_{\Theta |Z=z}^{\left(Q,\lambda \right)}}\left(\theta \right)$ in (Equation 73) can be interpreted as a log-likelihood ratio in a hypothesis test in which the objective is to distinguish the probability measures $P_{\Theta |Z=z}$ and $P_{\Theta |Z=z}^{\left(Q,\lambda \right)}$ based on the observation of the model θ. The former represents the algorithm under study trained upon z, whereas the latter represents a Gibbs algorithm trained upon the same dataset z.

From this perspective, the difference between the last two terms in (Equation 73), i.e.,(74)$$\begin{array}{c} \lambda \int \;\;\int \log\frac{dP_{\Theta |Z=z}}{dP_{\Theta |Z=z}^{\left(Q,\lambda \right)}}\left(\theta \right)dP_{\Theta }\left(\theta \right)dP_{Z}\left(z\right)\;-\;\lambda \int \;\;\int \log\frac{dP_{\Theta |Z=z}}{dP_{\Theta |Z=z}^{\left(Q,\lambda \right)}}\left(\theta \right)dP_{\Theta |Z=z}\left(\theta \right)dP_{Z}\left(z\right), \end{array}$$
can be interpreted as the variation of the expectation of the log-likelihood ratio(75)$$\begin{array}{c} \log\frac{dP_{\Theta |Z=z}}{dP_{\Theta |Z=z}^{\left(Q,\lambda \right)}}\left(\theta \right) \end{array}$$
when the probability measure from which the model θ and dataset z are drawn changes from the ground-truth distribution $P_{\Theta |Z}P_{Z}$ to the product of the corresponding marginals $P_{\Theta }P_{Z}$. As originally suggested in [10], Theorem 5 establishes an interesting connection between hypothesis testing, information measures, and generalization error. Nonetheless, this connection goes beyond this application in statistical machine learning as the same connection can be established directly from Theorem 3. This establishes a connection between the variation of the expectation due to changes in the probability measure, information measures, and hypothesis testing.

## 7. Final Remarks

A simple reformulation of Varadhan’s variational representation of relative entropy (Lemma 2) yields an explicit expression for the variation of the expectation of a real function when the probability measure shifts from a Gibbs measure to an arbitrary probability measure (Lemma 3). This result connects directly with information-projection methods, Pythagorean identities for relative entropy, and optimization problems with a constraint on the relative entropy of the measure to be optimized with respect to a reference measure (Lemma 4). A simple algebraic manipulation of Lemma 3 provides a general formula. It describes how the expectation changes with respect to changes in the probability measure (Theorem 1). The result is general, and the only assumption is that both initial (before the variation) and final (after the variation) measures are absolutely continuous with respect to a common reference. A key insight is the central role of Gibbs measures in this framework. The change in expectation is described through relative entropy comparisons between the initial and final measures, each with respect to a specific Gibbs measure built from the function under study. Notably, the reference measure of these Gibbs distributions need not be a probability measure. It can be a σ-finite measure, such as the Lebesgue measure or the counting measure. In such cases, the resulting expressions include Shannon’s fundamental information measures, including entropy and conditional entropy (Corollary 2). Building on these results, the variation of expectations under changes in joint probability measures is studied. Two cases are of special interest. In the former, one marginal remains unchanged (Theorem 2). In the latter, the joint measure changes to the product of its marginals (Theorems 3 and 4). In the case of Gibbs joint measures, the resulting expressions involve only standard information quantities: mutual information, lautum information, and relative entropy. These results show a broad connection between the variation in the expectation of measurable functions, induced by changes in probability measure, and information measures such as mutual and lautum information. 

## Figures and Tables

**Figure 1 entropy-27-00865-f001:**
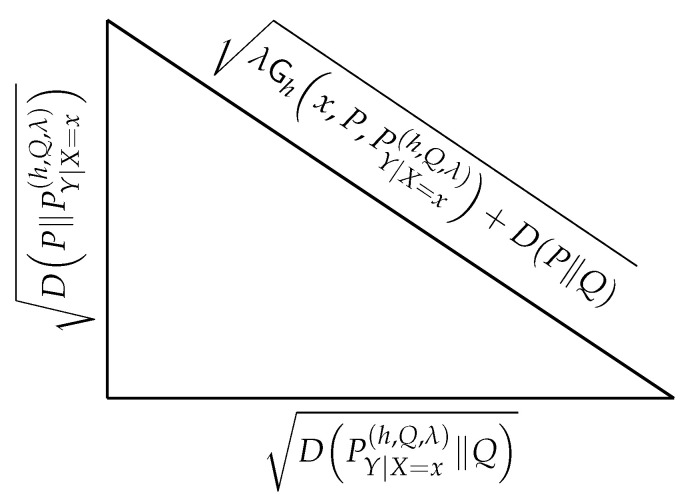
Geometric interpretation of Lemma 3, with *Q* a probability measure.

## Data Availability

No new data was generated; the article describes entirely theoretical research.

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
