# Peer review of "Variations on the Expectation Due to Changes in the Probability Measure"

_entropy, 2025, doi:10.3390/e27080865_

Round 1

Reviewer 1 Report

Comments and Suggestions for Authors

The main novelty of this work lies in providing closed-form expressions for the variation of expectations under measure changes,  framed through information-theoretic quantities. The mathematical treatment demonstrates both rigor and precision.

The application of Gibbs measures over sigma-finite references (including Lebesgue and counting measures) is sound. However, some general readers would benefit from clearer distinctions between measure-theoretic and probabilistic contexts. Additionally, several paragraphs are overly dense and lengthy, hindering the clarity of exposition.

Despite the quality of the work, the results remain predominantly theoretical, with only abstract applications discussed. The work lacks numerical examples, simulations, or real-world case studies that would demonstrate practical relevance and impact. Consequently, the presentation assumes readers possess advanced knowledge in probability theory and information theory, significantly limiting broader accessibility.

Finally, I recommend the authors to move most of the mathematical machinery of sections 3 and 4 to the appendix, and to add an application/example section  to improve readability and interest.

Author Response

Responses are in the attached pdf file.

Reviewer 2 Report

Comments and Suggestions for Authors

Review of the paper “variations on the Expectation due to Changes in the Probability Measure

The paper addresses the theoretical aspect of the variation of the expectation of a given function due to changes in the probability measure. Authors obtain closed-form expression both in the general case and when the reference measure is the Lebesgue measure or the counting one. The key of their result is the involvement of Gibbs probability measures, mutual and latum information.

The contribution is purely theoretical, and it generalizes similar results obtained into the specific field of error of machine learning algorithms.

Most of the proofs are direct consequences of definitions and/or of theorems and lemmas already known in literature. Just two more technical proofs are detailed reported into the appendix.

The language style is typical of mathematical papers: very succinct and technical. The English and the presentation are of a good level.

Bibliographic references are appropriate and exhaustive.

The only main lack that I see is the complete absence of any practical example that could help the comprehension and increase the interest on the contribution.

The only typo that I found is at line 141 of page 8, where a (1-\lambda) should be replaced by (1-\alpha).

Author Response

Answers are in the attached pdf file.

Round 2

Reviewer 1 Report

Comments and Suggestions for Authors

The authors have implemented the suggested changes and I have no further comments.

Reviewer 2 Report

Comments and Suggestions for Authors

Authors have addressed my previous recommendations, particularly the need for an explicit example. In fact, the revised version includes an example section that shows the closed form for the change in expectation of the difference between test and training error in machine learning algorithms. In the comments of such an example, a proper argument is made for interpreting the generalization error as a variation in the empirical risk function.